# Mobile Gaming for Cognitive Health in Older Adults: A Scoping Review of App Store Applications

**DOI:** 10.3390/healthcare13080855

**Published:** 2025-04-09

**Authors:** Jiadong Yu, Eunie Jung, D.A. Bekerian, Chelsee Osback

**Affiliations:** 1California School of Professional Psychology, Alliant International University, Fresno, CA 93727, USA; dbekerian@alliant.edu (D.A.B.); chelsee.osback@alliant.edu (C.O.); 2Bernerd College, University of the Pacific, Stockton, CA 95122, USA; ejung@pacific.edu

**Keywords:** cognitive training, mobile applications, older adults, cognitive decline prevention, digital health accessibility

## Abstract

Background: Mobile gaming applications are increasingly marketed as cognitive training tools for older adults, yet their scientific validity and accessibility remain uncertain. This scoping review evaluates their effectiveness and inclusivity. Methods: A systematic search of the Apple App Store and Google Play Store identified 227 applications, with 14 meeting inclusion criteria. Apps were assessed for scientific validity, theoretical foundation, accessibility, cognitive targeting, user engagement, and monetization models. Results: While all 14 apps claimed cognitive benefits, only one cited empirical research. None included baseline cognitive assessments or progress tracking. Accessibility was limited—eight apps had visual accommodations, but none provided auditory support. Six apps were English-only, restricting linguistic inclusivity. Monetization varied, with eight requiring in-app purchases or subscriptions, posing financial barriers. Conclusions: This review highlights critical gaps in the current cognitive gaming application market for older adults. Despite their popularity, cognitive training apps for older adults lack scientific validation and accessibility, limiting their effectiveness as cognitive interventions. Developers should integrate evidence-based training, adaptive assessments, and inclusive accessibility features such as voice guidance and multilingual support. Future research should prioritize longitudinal studies to assess real-world efficacy, refine interventions targeting memory, executive function, and processing speed, and enhance inclusive design for diverse aging populations.

## 1. Introduction

The aging population, known collectively as “baby boomers”, presents a particular problem for health services. Advancements in medical research and healthcare have enabled people to live longer, contributing to a dramatic demographic shift. According to the World Health Organization, by 2050, the number of people over 60 years old worldwide is projected to double [1]. However, with this growing aging population, the frequencies of certain age-related conditions, such as Alzheimer’s disease and dementia, are expected to rise [2]. Neurodegenerative diseases become far more common as one gets older; and the subsequent cost is expected to strain health care systems [2]. A recent survey identified preventing cognitive decline and dementia as a pressing health concern for older adults, prompting the World Health Organization to recognize it as a global mental health priority [1].

The significance of cognitive health in older adults extends beyond individual well-being, bearing substantial societal and economic implications. Cognitive decline and dementia are associated with increased healthcare costs, greater demands for caregiving resources, and reduced economic productivity due to early retirement and caregiver absenteeism. Consequently, enhancing cognitive health through effective interventions could significantly alleviate healthcare burdens and economic strain, emphasizing the societal value of preventive strategies. While aging is associated with a decline in cognitive function, dementia and cognitive impairments are not inevitable aspects of aging [3,4]. Several strategies have been identified as potentially mitigating many risk factors for dementia or cognitive decline [5,6]. These include non-pharmacological interventions such as cognitive training, which may delay dementia-related functional impairment [7,8]. While evidence supports the effectiveness of these approaches, they often require in-person participation, which can be inaccessible due to mobility issues, financial constraints, or transportation barriers. Additionally, many traditional interventions struggle to maintain long-term engagement, limiting their effectiveness [9,10]. Given these challenges, researchers have turned to technology-based interventions as more accessible and engaging alternatives.

Recent research suggests that neurodegeneration may be slowed and even countered through targeted interventions [11]. One emerging therapeutic approach is cognitive training, which involves practicing tasks targeting selective attention, memory, verbal fluency, and general cognitive functioning [12]. Importantly, these training tasks are based, loosely, on solid empirical evidence, derived from studies in cognitive psychology. To be more engaging, the cognitive training usually is introduced through some type of “game”. For example, the individual may be asked to find cats in a picture crowded with common everyday items within a limited amount of time, to gain points. The argument is that practicing mental skills can be used to combat cognitive decline.

The rise of cognitive training games has coincided with the rapid expansion of smartphone accessibility and mobile application development. Over the past decade, smartphone use has grown significantly worldwide, leading to an unprecedented increase in mobile app development [13,14]. Among these, digital applications have emerged as a promising yet underexplored tool for improving cognitive health in older adults [14,15].

The marriage of mobile technology and cognitive training has resulted in a plethora of cognitive training games, all purporting to enhance cognitive functions. While some of these claims remain questionable, research has shown the significant benefits of cognitive training, including hippocampal regeneration in older adults [16]. Such findings highlight the potential of cognitive training games to contribute to reducing neurodegenerative diseases and cognitive decline in an aging population.

Despite the growing body of research examining cognitive training apps, significant gaps remain. While some studies indicate that these apps can be beneficial, inconsistencies in intervention protocols have led to varying results, making it difficult to compare findings and draw firm conclusions [17]. Additionally, research assessing the effectiveness of these apps in real-world settings is still limited, and few reviews have systematically evaluated the key advertised features of cognitive training apps [18].

Yu, Bekerian, and Osback [19] argued that the internet and other virtual platforms can be extremely effective platforms for people to receive support for mental health issues. However, certain criteria must be met to ensure their effectiveness. Apps should provide empirical evidence supporting their claims, be user-friendly, affordable, available in multiple languages, and be designed with usability features tailored for older adults. Additionally, apps should incorporate features that encourage sustained use. For instance, Headspace, a widely used mindfulness app, utilizes daily reminders, progress tracking, and gamification elements to enhance user engagement over time.

In this context, “scientific validity” refers to the degree to which an application’s cognitive benefits have been substantiated by empirical research, typically involving rigorous methodologies such as randomized controlled trials or longitudinal studies. “Evidence-based claims” pertain to explicit assertions made by developers regarding the cognitive benefits of their applications, supported by the scientific literature, research findings, or recognized theoretical frameworks. For example, an app claiming to improve memory should ideally reference studies demonstrating measurable improvements in memory outcomes following structured use of the app.

It remains unclear whether existing cognitive training apps for older adults meet these criteria. While some studies have examined mobile apps designed for cognitive training, few have systematically evaluated them using these essential parameters [19]. This omission is particularly concerning given the vulnerability of older adults to misleading claims, such as those suggesting that a given app can “prevent Alzheimer’s”. Such statements, when not backed by empirical research, pose significant ethical concerns, as older users may be misled into investing time and financial resources into ineffective interventions [19].

This study aims to address these gaps by conducting a scoping review of cognitive training games designed for older adults. Using the criteria established by Yu et al. [19], this review will assess mobile apps in terms of scientific validity, accessibility, usability, and motivation for sustained engagement. Additionally, it will explore how apps define and cater to the “elderly” population, as definitions vary widely, sometimes categorizing individuals over 55 as elderly, while in other cases, the threshold is set at 65 or older. Recognizing the heterogeneity within this population is critical, as cognitive needs may differ significantly between individuals aged 65 and those aged 85 and above.

The primary objective of this study is to contribute to the existing knowledge base by evaluating the current landscape of cognitive gaming applications for older adults. Specifically, this review seeks to answer the question: Are mobile apps designed to improve older adults’ cognitive functioning meeting consumer needs? To explore this question, the review will provide a detailed analysis of these apps across various domains, including theoretical foundations, evidence-based claims, accessibility features, and targeted cognitive functions. By identifying strengths and gaps in current app offerings, this study aims to provide insights that can guide future app development and ensure that cognitive training apps are both effective and ethically responsible.

The key questions that this review intends to answer are as follows:What scientific evidence supports the cognitive benefits claimed by these mobile applications?How accessible and usable are cognitive training apps for older adults?Do these apps effectively maintain user engagement?How do these applications cater to the diverse and heterogeneous needs of older populations?

## 2. Methods

This scoping review was conducted and reported in accordance with the 2018 Preferred Reporting Items for Systematic Reviews and Meta-Analyses extension for Scoping Reviews (PRISMA-ScR) guidelines [20]. This methodological approach ensures a comprehensive mapping of the existing landscape of cognitive game applications designed for older adults. The review aims to systematically identify, categorize, and evaluate these applications based on their theoretical foundations, accessibility, and user engagement features. 

### 2.1. Search Strategy and Eligibility Criteria

A systematic search was performed across the two most prominent mobile application distribution platforms, the Apple App Store (iOS 18, United States) and Google Play Store (Android 15, United States), to identify cognitive game applications targeted at older adults. The search employed a combination of keywords, including “Elders”, “Seniors”, “Older Adults”, “Memory”, “Attention”, “Cognitive”, and “Games”. The search algorithms of both platforms rank applications based on factors such as popularity and engagement, which influenced the order of retrieval.

Applications were considered eligible according to the following criteria: (1) Applications had to explicitly state that their primary focus was cognitive enhancement. (2) Applications that had been updated within the previous year were included to ensure that only actively maintained software was reviewed. (3) Only applications available in English were considered to facilitate standardization in assessment. (4) To ensure usability, the included applications were required to function as standalone applications without reliance on external hardware such as wearable monitors or web-based services requiring human support. (5) The only target audience was older adults.

Exclusion criteria were established to refine the selection process. Applications were excluded if they (1) were duplicates (if an app was available for both iOS and Android operating systems, we included the iOS version only); (2) had not received an update in the previous year; (3) had insufficient descriptive information in the app store listings that prevented meaningful evaluation.

### 2.2. Screening and Selection Process

The selection process was conducted in multiple stages to ensure rigorous appraisal and minimize bias. Initially, for each keyword searched, application titles and descriptions were reviewed to determine their relevance to cognitive enhancement in older adults. If the store description alone was insufficient to confirm eligibility, the application was downloaded for further assessment. The first round of screening was conducted independently by two researchers (J.Y. and D.A.B.). If disagreements arose, they were resolved through consensus discussions with a third and fourth researcher (E.J. and C.O.).

All applications deemed eligible underwent an independent feature assessment by two researchers. This assessment included reviewing the app’s theoretical foundation, whether it incorporated cognitive training principles, and if any claims of scientific validation were provided. Additionally, accessibility features, such as large text options, color contrast adjustments, and auditory feedback, were documented. Each application’s monetization model (free, in-app purchases, or subscription-based) was also recorded, along with its user engagement statistics, including the number of reviews and average rating. An adapted PRISMA flow diagram was used to document the entire selection process, ensuring transparency and replicability.

### 2.3. App Evaluation

The evaluation of the included applications focused on several key dimensions, including their theoretical foundation, evidence-based claims, accessibility features, targeted cognitive functions, user engagement metrics, and cultural inclusivity. Each application was systematically assessed based on descriptive and qualitative criteria extracted from the app store listings and additional in-app functionality testing. Theoretical foundation was assessed by examining whether the application referenced established cognitive training theories, such as working memory training, neuroplasticity principles, or problem-solving frameworks. Evidence-based claims were evaluated to determine if the app developers cited empirical studies or made general assertions about cognitive benefits without supporting evidence.

Accessibility features were reviewed to assess whether the app accommodated older adults with visual or hearing impairments. This included evaluating font size options, high-contrast modes, audio guidance, and adaptability of the user interface. Additionally, apps were assessed for language availability and inclusivity for diverse populations. Cultural inclusivity was assessed by examining whether the apps provided culturally relevant content and multilingual support beyond just English and whether they considered diverse ethnic and cultural contexts in their design and content.

The targeted cognitive functions of each app were categorized based on whether they aimed to enhance memory, attention, problem solving, reaction time, or general cognitive flexibility. Apps that provided structured cognitive exercises and real-time feedback mechanisms were distinguished from those that primarily functioned as casual games with implicit cognitive engagement. User engagement was evaluated using quantitative data extracted from the app store, including the number of reviews, average user ratings, and update frequency. Apps with a high volume of positive reviews were examined to determine whether user testimonials aligned with the developers’ cognitive enhancement claims. Applications that had not been updated within the past year were flagged as potentially obsolete. Monetization models were also recorded to assess whether apps were freely accessible, included in-app purchases, or required a subscription-based payment structure. Each of these dimensions was systematically documented to provide a comprehensive evaluation of the usability, credibility, and accessibility of cognitive game applications designed for older adults.

## 3. Results

In this section, we provide a summary of the search results, the selection process of the identified applications, and an overview of the primary focus areas used in evaluating the selected cognitive game applications. The results are organized into thematic subsections to systematically address the central evaluation criteria, including claims of cognitive benefits, scientific validation, targeted cognitive concerns, accessibility features, and user engagement metrics.

### 3.1. Study Selection and Characteristics

The search was conducted in January 2025, and the app selection process is illustrated in Figure 1. A total of 227 potentially relevant apps were identified from the App Store (*n* = 125) and Google Play (*n* = 102). After removing 100 duplicate records, 127 unique apps were screened based on their titles and descriptions. Of these, 113 apps were excluded due to irrelevance or failure to meet inclusion criteria. A total of 14 game applications designed to enhance cognitive function in older populations were included in this scoping review. Of these, three apps were retrieved from Google Play and 11 from the App Store. However, all 14 applications were available on both platforms.

As presented in Table 1, a central aspect of evaluating the cognitive game applications involved analyzing the claims regarding cognitive benefits and the extent to which these claims were supported by scientific evidence. Out of the 14 applications included in this review, 12 asserted that their games could enhance cognitive function without explicitly citing any supporting research or empirical validation. These applications provided general statements about improving memory, attention, or overall cognitive health but did not reference specific studies or methodologies validating their claims.

One application cited a randomized controlled trial (RCT) as evidence of its effectiveness. It also claimed that it had been endorsed by the Alzheimer’s Association, suggesting a level of credibility. However, upon further review, the application did not provide access to the study or further details about the methodology or findings of the RCT. Another application stated that it was developed based on 18 months of research but did not include references to specific studies, researchers, or institutions involved in the research process. This lack of explicit scientific support leaves the validity of its cognitive benefits unverified.

Additionally, none of the 14 applications reviewed provided information on population norms. There was no evidence indicating that these applications were normed specifically for older adults or tested within a population of older users to determine their efficacy for age-related cognitive decline. Furthermore, none of the reviewed applications mentioned any theoretical framework or established cognitive science principles that guided their game design.

### 3.2. Screening, Targeted Cognitive Concerns, and Progress Assessment

None of the reviewed applications implemented any form of cognitive screening to assess the user’s baseline cognitive abilities before engaging in gameplay. Despite making claims about cognitive benefits, these applications did not include initial cognitive assessments to determine the suitability of the game for users with varying cognitive conditions. Similarly, none of the applications provided a structured method for tracking cognitive progress over time. While all 14 applications claimed to support cognitive improvement, there were no embedded features that allowed users to monitor their cognitive changes, assess their progress, or receive feedback on their cognitive development.

Regarding the targeted cognitive concerns of these applications, there was considerable variability in their stated objectives. Four applications primarily emphasized relaxation, engagement, and enjoyment rather than explicitly aiming to enhance cognitive abilities. Two applications focused on cognitive stimulation but lacked clarity on which specific cognitive processes were being trained. One application addressed accessibility for seniors while incorporating cognitive stimulation, ease of use, and relaxation. Three applications explicitly targeted short-term memory, communication skills, and caregiver engagement, suggesting a potential emphasis on supporting cognitive and social functioning in aging populations. The remaining four applications claimed to support various cognitive functions, including memory training, hand–eye coordination, social engagement, mental agility, vocabulary enhancement, problem solving, spatial reasoning, mathematical reasoning, and logical problem solving.

### 3.3. Accessibility, Cultural Inclusivity, and Age Considerations

Accessibility features varied across the reviewed applications, with a predominant focus on vision accessibility while auditory accommodation remained largely absent. Eight applications incorporated enlarged visual elements, such as large puzzle pieces, oversized buttons, and high-contrast text to improve usability for individuals with visual impairments. Three applications specifically included zoom functions and adjustable text sizes to enhance readability. However, none of the applications explicitly mentioned auditory accommodation, such as voice-guided navigation, text-to-speech functionalities, or auditory feedback for users with hearing impairments. Two applications featured customizable visual themes and color adjustments, allowing users to modify contrast settings to suit their visual preferences. Several applications promoted a senior-friendly interface, offering intuitive controls and minimal navigation complexity, yet the degree to which these interfaces were optimized for users with limited technological proficiency varied considerably.

Language accessibility was another area of variability among the reviewed applications. Six of the 14 applications were only available in English, offering no alternative language options. The remaining applications supported multiple languages, but the extent of linguistic inclusivity varied significantly. One application was available in only two languages—English and Indonesian, whereas four applications provided support for up to 59 different languages. In terms of cultural inclusivity, none of the applications explicitly incorporated ethnicity- or religion-based adaptations to cater to diverse user backgrounds. There were no indications that the games were designed with cultural sensitivity in mind, nor was there any mention of localized content that reflected the values, traditions, or lived experiences of different ethnic or religious groups.

The definition of older adults as a target audience was also vague across the reviewed applications. All 14 applications were marketed toward “seniors”. Yet none provided a clear age-based classification to differentiate between younger seniors (e.g., 60–70 years old) and older populations (e.g., 80+ years old).

### 3.4. Pricing, Platform Availability, and User Engagement

Among all the reviewed applications as demonstrated in Table 2, only one required an internet connection. The remaining 13 could be used offline. In terms of platform availability, three applications were accessible on both mobile and tablet devices, and one application was exclusively available on tablets. The remaining nine applications were compatible across multiple platforms, i.e., mobile, tablet, and laptop platforms.

The number of user reviews among the reviewed applications ranged from 3 to 605,085. Two applications had review counts exceeding 390,000, while one had the highest number of reviews, at 605,085. Four applications had between 30,000 and 45,000 reviews, while another four had between 8000 and 14,000 reviews. Three applications had significantly fewer reviews, including 3, 5, and 25 reviews.

Most applications had high ratings, with 11 out of 14 receiving 4.7 out of 5.0 or higher. There was a range in ratings. Two applications received the highest rating of 4.9 out of 5.0, while others had slightly lower ratings, such as 4.2 or 4.3 out of 5.0.

The update frequency also varied across applications. Three applications had been updated in January 2025, while two were last updated in February 2024. Five applications had their last updates between April and September 2024, while four applications had their most recent updates between October and December 2024.

The pricing structures of the reviewed applications varied widely, with six applications offering entirely free access, while the remaining eight incorporated in-app purchases or subscription-based pricing. Subscription-based models were present in four applications, with pricing structures ranging from short-term monthly subscriptions to annual membership plans. One application offered a 14-day free trial, followed by a subscription costing USD 9.99 per month or USD 69.99 per year. Another application provided a one-week free trial, after which users were charged USD 6.00 per month for individual use or USD 9.00 per month per tablet for professional access.

Other applications featured various in-app purchases, allowing users to remove advertisements or purchase virtual currency for gameplay enhancements. The cost of these purchases similarly differed. Four applications offered ad removal for USD 5.99 to 10.99, while in-app currency purchases ranged from USD 0.99 to 19.99, depending on the bundle size.

Four applications followed a hybrid pricing model, offering free gameplay with the option to purchase additional content. One application allowed users to subscribe to a Weekly Subscription for USD 1.99, a Monthly Subscription for USD 4.99, or a Yearly Subscription for USD 14.99. Another application followed a similar model, with a Monthly Subscription priced at USD 1.99, while also allowing users to remove ads for USD 10.99 or purchase various in-game currency bundles ranging from USD 0.99 to 3.99. A separate application provided a Mega Bundle for USD 12.99, a Super Bundle for USD 5.99, and additional small purchases for in-game content.

## 4. Discussion

This scoping review addresses several critical gaps in the current landscape of cognitive gaming applications for older adults by explicitly responding to the key questions posed in the Introduction. Regarding our first research question, while all reviewed applications claimed to enhance cognitive function, there was a widespread lack of scientific validation, population norms, and theoretical frameworks to support these claims. The absence of empirical evidence calls into question whether these applications genuinely contribute to cognitive improvement or merely serve as casual entertainment tools marketed toward older users. Previous research has emphasized the importance of empirical validation in cognitive training interventions [21,22]. Without validation through peer-reviewed research or clinical trials, users and healthcare professionals have no way to assess the credibility or efficacy of these cognitive interventions.

A critical gap identified was the absence of cognitive screening and progress tracking within the reviewed applications. Prior studies have demonstrated that baseline assessments and progress tracking are essential for evaluating cognitive improvement [23]. None of the applications included structured mechanisms to assess the baseline cognitive abilities of users or track improvements over time. This absence of assessment tools makes it difficult to determine whether users experience tangible cognitive enhancements or whether any observed benefits are incidental rather than systematic. Without objective measures, the effectiveness of these applications remains uncertain.

The targeted cognitive concerns varied across applications, with some focusing on general relaxation and entertainment and others addressing specific cognitive functions such as memory, problem solving, and mental agility. However, cognitive training research suggests that standardized training methodologies, such as spaced repetition, dual-task training, and executive function exercises, are necessary for measurable cognitive gains [24,25]. The absence of standardized cognitive training methodologies and validated interventions suggests that these applications may not be intentionally designed to provide targeted cognitive benefits.

Addressing our second research question, we found that accessibility features were inconsistent across applications. While there was evidence for applications prioritizing vision accessibility, auditory accommodation was very limited or absent. No applications included voice-guided navigation, text-to-speech functionalities, or auditory cues to assist hearing-impaired users. This lack of auditory support limits accessibility for older adults with hearing impairments, who may struggle to engage with applications that rely solely on visual cues. Developers should incorporate auditory accessibility features such as customizable volume controls, real-time voice guidance, and text-to-speech functionalities to enhance usability for hearing-impaired users.

Language accessibility was also limited, with several applications available exclusively in English, potentially restricting access for non-English-speaking users. While some applications offered multiple language options, the range varied significantly. Given the linguistic diversity among older adults, broader language support and integrated culturally relevant content can enhance user engagement across different regions.

Additionally, none of the applications explicitly incorporated cultural or religious inclusivity, and no specific adaptations were made for diverse ethnic backgrounds. Cultural relevance plays a crucial role in user engagement, especially for older adults who may feel disconnected from generic, Western-centric cognitive training applications. Applications should incorporate culturally adapted content, such as region-specific themes, multilingual voice support, and culturally familiar game elements, to increase accessibility and engagement among diverse aging populations.

In relation to our third research questions, platform availability and internet dependence were not major barriers, as nearly all applications do not require an internet connection. However, there were notable disparities in updating frequency and developer engagement. Some applications had been updated as recently as a few months prior to the data collection, while others had not received updates for over a year. Applications that are not actively maintained may become obsolete or fail to deliver long-term cognitive benefits to users. A lack of regular updates may also impact security and usability, particularly as mobile operating systems evolve and introduce new accessibility features. Implementing structured update cycles can ensure that applications remain compatible with the latest devices and incorporate emerging best practices in user interface design for older adults. Developers should also consider transparency regarding update schedules, informing users about planned improvements and addressing feature requests based on user feedback.

Considering the final research question, another major concern was the lack of age-specific classifications among the reviewed applications. While all applications were marketed toward seniors, none differentiated between younger seniors and older populations. Cognitive training needs and capabilities can vary significantly within the aging population [26], yet these applications treated all older adults as a homogeneous group. Research has shown that cognitive decline manifests differently across age brackets, with older seniors (e.g., 80+ years) requiring different intervention strategies compared to younger seniors (e.g., 60–70 years) [27]. Customizable cognitive training programs that consider age-related differences in cognitive decline, physical limitations, and learning preferences would optimize engagement and effectiveness.

The pricing models of these applications varied widely, with several applications also including microtransactions for virtual currency or ad removal. Financial barriers can hinder the adoption of cognitive training interventions, particularly among older adults with fixed incomes [28]. Research suggests that pricing transparency and alternative funding models, such as subsidized access through healthcare programs, can enhance affordability and user engagement [29]. Given the potential benefits of cognitive training for aging populations, more applications should consider offering fully accessible versions with essential features at no cost or incorporating non-monetary reward systems to enhance user engagement without financial constraints. A balance between financial sustainability and accessibility is necessary to ensure that these applications remain viable while still being inclusive for older adults.

This review has several limitations. First, it only included applications available in the Apple App Store and Google Play Store. Other cognitive training applications distributed through alternative platforms, such as specialized health software or web-based training programs, were likely omitted. Additionally, this review lacked empirical usability testing with older adults to observe real-world usability, engagement, and effectiveness. Although in-app functionality was assessed, direct observation, usability testing, and qualitative feedback from older adults would have provided deeper insights. Furthermore, longitudinal or follow-up studies to evaluate sustained cognitive benefits and long-term effectiveness were beyond the scope of this scoping review.

Moreover, although user engagement metrics such as app ratings and reviews were recorded, these data were obtained directly from app stores, limiting the ability to independently verify or deeply analyze their accuracy. App store ratings and reviews are also susceptible to biases, including promotional efforts by developers and selection bias from early adopters. Future research should aim to include independent and comprehensive analyses of user engagement, adherence rates, and qualitative assessments to evaluate user satisfaction, ease of use, and barriers to sustained usage more accurately.

None of the reviewed applications provided strong empirical validation of their cognitive benefits. The absence of RCTs and peer-reviewed clinical studies supporting these applications raises concerns about their credibility. Future research should prioritize rigorous experimental methodologies, including longitudinal studies tracking cognitive outcomes over time. Developers should integrate scientific validation by collaborating with cognitive scientists, neuroscientists, and gerontologists. Applications should undergo thorough clinical or research validation before being marketed for cognitive enhancement. Additionally, baseline cognitive assessments and dynamic progress-tracking features should be implemented to enhance usability and credibility.

Additionally, greater attention should be given to accessibility, particularly for hearing-impaired users, and efforts should be made to enhance linguistic and cultural inclusivity. The expansion of language options, as well as culturally adapted content, could significantly improve accessibility and engagement for diverse aging populations. Furthermore, future applications should differentiate between different age groups within the senior population and tailor their cognitive training programs accordingly, recognizing the variability in cognitive needs and abilities across different age brackets.

## 5. Conclusions

While cognitive training applications for older adults hold promise, the current landscape is characterized by a lack of scientific rigor, accessibility challenges, and limited empirical validation. Most applications claim to enhance cognitive function but lack empirical evidence to substantiate their claims, raising concerns about their effectiveness and ethical implications. Additionally, the absence of standardized training methodologies, cognitive screening tools, and progress-tracking mechanisms further limits their potential as reliable cognitive interventions.

Future efforts should prioritize integrating evidence-based cognitive training methodologies, including structured exercises based on validated theoretical frameworks such as dual-task paradigms, spaced repetition, and working memory models. Ensuring inclusivity through auditory and visual accessibility enhancements, multilingual support, and culturally adapted content is crucial. Specific usability testing methodologies that are recommended include user-centered design approaches, cognitive walkthroughs, and field usability testing, with direct observations and feedback from older adult users.

Developers should collaborate with cognitive scientists and healthcare professionals to incorporate validated cognitive exercises, improve accessibility for diverse populations, and provide transparent information regarding the effectiveness of their applications. Moreover, future research should focus on conducting longitudinal studies to assess the real-world impact of these applications and determine their long-term benefits for cognitive health. By addressing these gaps, cognitive training applications can evolve into credible and effective tools for promoting cognitive well-being in older adults, ultimately contributing to healthier aging and improved quality of life.

## Figures and Tables

**Figure 1 healthcare-13-00855-f001:**
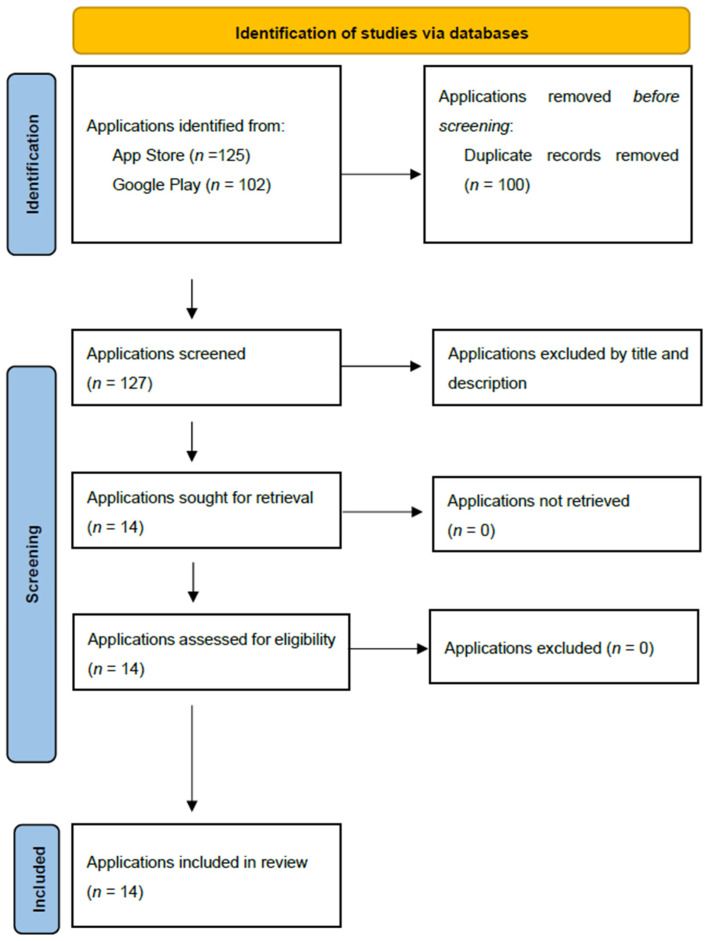
PRISMA flow diagram of the scoping review process: Claims of cognitive benefits and scientific basis.

**Table 1 healthcare-13-00855-t001:** Scientific basis, theoretical framework, and cognitive targeting of apps.

App Name	Evidence-Based	Population-Specific Norming	Theory	Screening	Targeted Concerns	Assessment of Progress	Internet Dependence
Vita Jigsaw for Seniors	No claims of research-based validation	No	lacks explicit theoretical foundation	No formal screening	Relaxation, fun, and joy	No	No
Solitaire for Seniors Game	Claims cognitive benefits, but no mention of research backing	No	lacks explicit theoretical foundation	No formal screening	Addresses accessibility for seniors, cognitive stimulation, ease of use, and relaxation	No	No
Oya: Alzheimer Game, Match	Claims to help slow down Alzheimer’s progression, but no research cited	No	lacks explicit theoretical foundation	No formal screening	Targets cognitive function, especially short-term memory	No	No
Memory Lane Games	RCT evidence cited, endorsed by Alzheimer’s associations	No	lacks explicit theoretical foundation	No formal screening	Targets cognitive function, communication, and caregiver engagement	No	No
SCARLETT, your memory coach	Designed with health professionals, no explicit research study mentioned	No	lacks explicit theoretical foundation	Includes professional follow-up and tracking for patients	Targets cognitive function, memory training, and engagement	No	No
Reshuffle App	No explicit scientific study mentioned	No	lacks explicit theoretical foundation	No formal screening	Targets cognitive function, memory, hand-eye coordination, and social engagement	No	Yes
Vita Word Search for Seniors	Claims cognitive benefits, but no explicit research study mentioned	No	lacks explicit theoretical foundation	No formal screening	Targets cognitive function, mental agility, and vocabulary enhancement	No	No
Vita Block for Seniors	Claims cognitive benefits, but no explicit research study mentioned	No	lacks explicit theoretical foundation	No formal screening	Targets cognitive function, memory, problem solving, and spatial reasoning	No	No
Vita Math Puzzle for Seniors	Claims cognitive benefits, but no explicit research study mentioned	No	lacks explicit theoretical foundation	No formal screening	Targets mathematical reasoning, logical problem solving, and mental agility	No	No
Vita Sudoku for Seniors	Claims cognitive benefits, but no explicit research study mentioned	No	lacks explicit theoretical foundation	No formal screening	Targets logical reasoning, problem solving, and mental agility	No	No
Vita FreeCell for Seniors	Claims cognitive benefits, but no explicit research study mentioned	No	lacks explicit theoretical foundation	No formal screening	Targets mental sharpness, problem solving, and cognitive stimulation	No	No
Vita Solitaire for seniors	Claims cognitive benefits, but no explicit research study mentioned	No	lacks explicit theoretical foundation	No formal screening	Targets mental sharpness, problem solving, and cognitive stimulation	No	No
Vita Crossword—Word Games	Claims cognitive benefits, but no explicit research study mentioned	No	lacks explicit theoretical foundation	No formal screening	Targets memory recall, cognitive flexibility, and mental agility	No	No
Vita Mahjong	Claims cognitive benefits, but no explicit research study mentioned	No	lacks explicit theoretical foundation	No formal screening	Targets memory, pattern recognition and cognitive engagement	No	No

**Table 2 healthcare-13-00855-t002:** Accessibility, pricing, platform availability, and user engagement of mobile apps.

App Name	Platform	Pricing Model	Number of Languages	Accessibility	Ethnicity and Religion	Age Specific After 65+	Number of Reviews	Ratings	Last Update Date
Vita Jigsaw for Seniors	Mobile/Tablet	Free	1	Vision-friendly features	No explicit mention	No	44,688	4.7/5	4 September 2024
Solitaire for Seniors Game	Mobile/Tablet/Laptop	Free with in-app purchases	1	Vision-friendly features	No explicit mention	No	44,048	4.9/5	6 January 2025
Oya: Alzheimer Game, Match	Mobile/Tablet	Free	1	Vision-friendly features	No explicit mention	No	222	4.3/5	22 November 2024
Memory Lane Games	Mobile/Tablet	14-day free trial then Subscription	1	No explicit mention	No explicit mention	No	25	4.5/5	21 January 2025
SCARLETT, your memory coach	Tablet	Free for one week, then Subscription	1	No explicit mention	No explicit mention	No	5	4.2/5	12 February 2024
Reshuffle App	Mobile/Tablet/Laptop	Free	1	Vision-friendly features	No explicit mention	No	3	5.0/5.0	28 January 2025
Vita Word Search for Seniors	Mobile/Tablet/Laptop	Free with in-app purchases	3	Vision-friendly features	No explicit mention	No	43,491	4.7/5.0	10 July 2024
Vita Block for Seniors	Mobile/Tablet/Laptop	Free	5	Vision-friendly features	No explicit mention	No	37,179	4.7/5.0	11 June 2024
Vita Math Puzzle for Seniors	Mobile/Tablet/Laptop	Free with in-app purchases	59	Vision-friendly features	No explicit mention	No	13,751	4.7/5.0	26 April 2024
Vita Sudoku for Seniors	Mobile/Tablet/Laptop	Free with in-app purchases	58	Vision-friendly features	No explicit mention	No	8747	4.7/5.0	19 April 2024
Vita FreeCell for Seniors	Mobile/Tablet/Laptop	Free with in-app purchases	59	Vision-friendly features	No explicit mention	No	9762	4.7/5.0	1 July 2024
Vita Solitaire for seniors	Mobile/Tablet/Laptop	Free with in-app purchases	59	Vision-friendly features	No explicit mention	No	391,612	4.7/5.0	23 January 2025
Vita Crossword—Word Games	Mobile/Tablet/Laptop	Free with in-app purchases	2	Vision-friendly features	No explicit mention	No	31,353	4.7/5.0	9 August 2024
Vita Mahjong	Mobile/Tablet/Laptop	Free with in-app purchases	59	Vision-friendly features	No explicit mention	No	605,085	4.9/5.0	23 January 2025

## Data Availability

The original contributions presented in this study are included in the article. Further inquiries can be directed to the corresponding author(s).

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
