# Peer review of "Mobile Gaming for Cognitive Health in Older Adults: A Scoping Review of App Store Applications"

_healthcare, 2025, doi:10.3390/healthcare13080855_

Round 1

Reviewer 1 Report

Comments and Suggestions for Authors

I would like to exoress my gratitude for the invitation to review the manuscript. Upon careful reading oof the article, I find the research topic to be intriguing. Nevertheless, I have several recommendations that could enhance the quality of the paper.

ABSTRACT

Include more details about the search strategy and selection process to provide a clearer understanding of how applications were evaluated.

Incorporate quantitative data or specific examples to illustrate the extent of issues like lack of scientific validation or accessibility features.

Elaborate on the implications of these findings for stakeholders, such as developers, healthcare providers, and policymakers, to provide a more comprehensive perspective.

Offer more specific suggestions for future research, such as particular cognitive functions that should be targeted or innovative accessibility features that could be developed.

INTRODUCTION

Expanding on why cognitive health in older adults is an important area of study could enhance the Introduction. Discussing the potential societal or economic impacts of improved cognitive health through mobile gaming would provide additional context and underscore the relevance of the research.

The Introduction could clarify what is meant by "scientific validity" and "evidence-based claims" in the context of app evaluation. Providing definitions or examples would ensure that readers understand these critical evaluation criteria.

METHODS

While the methodology mentions assessing the theoretical foundation of applications, it could benefit from more explicit criteria or examples of what constitutes a strong theoretical basis for cognitive training apps.

Although user engagement statistics are recorded, a deeper analysis or comparison of these metrics across different applications could provide more insights into their popularity and effectiveness.

Consideration of cultural diversity in app design and content could be addressed, given that cognitive health needs might vary across different cultural contexts.

While the methodology mentions in-app functionality testing, it could be enhanced by including empirical testing with older adults to observe real-world usability and effectiveness.

Incorporating longitudinal data or follow-up studies on user outcomes could provide deeper insights into the long-term cognitive benefits of these apps.

Including a comparative analysis between different types of apps (e.g., those with structured exercises vs. casual games) could offer more detailed insights into what features are most beneficial for cognitive health.

DISCUSSION

While recommendations are made, they could be more specific. For instance, suggesting particular methodologies for usability testing or examples of successful collaborations between developers and scientists could offer clearer guidance.

Author Response

Dear Reviewer

We sincerely appreciate the time and effort you have dedicated to reviewing our manuscript. Your comments and suggestions have been invaluable in improving the quality and clarity of our work. Below, we provide a detailed, point-by-point response to address each of your comments and suggestions.

Thank you again for your thoughtful feedback.

Reviewer's Comment

ABSTRACT

Include more details about the search strategy and selection process to provide a clearer understanding of how applications were evaluated. Incorporate quantitative data or specific examples to illustrate the extent of issues like lack of scientific validation or accessibility features. Elaborate on the implications of these findings for stakeholders, such as developers, healthcare providers, and policymakers, to provide a more comprehensive perspective. Offer more specific suggestions for future research, such as particular cognitive functions that should be targeted or innovative accessibility features that could be developed.

Response: Thank you for these valuable suggestions. We have revised the abstract to address the comments, incorporating details on the search strategy, selection process, quantitative findings, stakeholder implications, and future research directions.

INTRODUCTION

Expanding on why cognitive health in older adults is an important area of study could enhance the Introduction. Discussing the potential societal or economic impacts of improved cognitive health through mobile gaming would provide additional context and underscore the relevance of the research. The Introduction could clarify what is meant by "scientific validity" and "evidence-based claims" in the context of app evaluation. Providing definitions or examples would ensure that readers understand these critical evaluation criteria.

Response: Thank you for these valuable suggestions. We have revised the introduction to explicitly highlight the societal and economic impacts of cognitive health on older adults. Additionally, we clarified critical evaluation criteria. "Scientific validity" is now clearly defined as the extent to which cognitive benefits claimed by applications are supported by empirical research. Similarly, we clarified "evidence-based claims" by providing examples, emphasizing that these claims should be directly backed by scientific studies demonstrating measurable outcomes.

METHODS

While the methodology mentions assessing the theoretical foundation of applications, it could benefit from more explicit criteria or examples of what constitutes a strong theoretical basis for cognitive training apps.

Response: We appreciate your suggestion and have revised the methodology section to explicitly specify criteria for evaluating the theoretical foundation of cognitive training applications.

Although user engagement statistics are recorded, a deeper analysis or comparison of these metrics across different applications could provide more insights into their popularity and effectiveness.

Response: We acknowledge the importance of a comparative analysis of user engagement metrics. However, since the engagement statistics were provided by the platforms themselves and we do not have access to independently verify or further analyze these data, we have clearly noted this limitation in the manuscript.

Consideration of cultural diversity in app design and content could be addressed, given that cognitive health needs might vary across different cultural contexts.

Response: We agree with this valuable point and have revised the methodology and evaluation sections to explicitly include an assessment criterion addressing cultural diversity and inclusivity.

While the methodology mentions in-app functionality testing, it could be enhanced by including empirical testing with older adults to observe real-world usability and effectiveness. Incorporating longitudinal data or follow-up studies on user outcomes could provide deeper insights into the long-term cognitive benefits of these apps.

Response: We recognize the importance of empirical usability testing and longitudinal studies; however, collecting data related to real-world usability, effectiveness, and long-term outcomes was beyond the scope of this scoping review. We have explicitly acknowledged these limitations in the manuscript and recommended them as important areas for future research.

Including a comparative analysis between different types of apps (e.g., those with structured exercises vs. casual games) could offer more detailed insights into what features are most beneficial for cognitive health.

Response: We appreciate your valuable suggestion regarding a comparative analysis. While our scoping review specifically focuses on mobile gaming applications claiming cognitive benefits, one central aim was to evaluate whether these claims are scientifically validated. Given this aim, our intent was not to conduct an in-depth comparative analysis of different types of apps, but rather to broadly identify and evaluate whether apps meet established scientific and accessibility criteria. Although differentiating between structured cognitive training apps and casual gaming apps was beyond the current review’s scope, we acknowledge that such a comparative analysis would indeed offer valuable insights into which specific features are most beneficial. We have explicitly mentioned this as a limitation in our manuscript and highlighted it as an important direction for future research.

DISCUSSION

While recommendations are made, they could be more specific. For instance, suggesting particular methodologies for usability testing or examples of successful collaborations between developers and scientists could offer clearer guidance.

Response: Thank you for your valuable feedback. We have now revised the conclusion by explicitly suggesting particular methodologies for usability testing, such as cognitive walkthroughs, user-centered design approaches, and field usability testing involving direct observation of older adults to demonstrate concrete pathways for effective app development.

Reviewer 2 Report

Comments and Suggestions for Authors

The manuscript addresses cognitive health in older adults through mobile gaming applications, with a particular focus on cognitive training. It follows a systematic scoping review methodology (PRISMA) and aims to identify critical gaps in the scientific validity, accessibility, and effectiveness of mobile applications designed for cognitive enhancement.

There are both strengths and weaknesses in the manuscript that should be highlighted.

Strengths:

  • The manuscript adheres to PRISMA guidelines and includes a well-documented scoping review methodology, offering an understanding of the current landscape for the relevant mobile gaming apps.
  • The discussion outlines the advantages and limitations of existing applications from a mental health perspective in older adults.

Weaknesses:

  • The abstract mentions a unique application that cites empirical evidence, yet there is no in-depth evaluation of this app within the manuscript. A comparative analysis of its empirical results against the strengths and weaknesses of other applications would provide a more comprehensive perspective on successful mobile gaming applications.
  • Table 1 reveals that none of the reviewed applications incorporate evidence-based cognitive training methodologies or assessment mechanisms to track cognitive progress. Given this, it would be beneficial to include a dedicated section—or at least a discussion paragraph—exploring cognitive enhancement methodologies from a healthcare perspective and evaluating the applications accordingly.
  • All of the applications lack an explicit theoretical foundation. Offering recommendations for integrating established cognitive science principles into app development would significantly enhance the quality of the paper.
  • The study primarily focuses on general usability features of the applications but does not sufficiently compare them against existing cognitive training interventions, whether digital or traditional. Strengthening the Related Work section by linking traditional cognitive training research to digital approaches would provide valuable context.
  • While app store ratings and user reviews are included in the analysis, these metrics are susceptible to bias, such as promotional manipulation. The manuscript would benefit from incorporating usability studies to provide a more detailed assessment of user engagement.

While the study is interesting and addresses a relevant topic, several critical points require further development, particularly regarding empirical evaluation, in-depth analysis of the selected applications, and stronger connections between cognitive health methodologies and mobile gaming apps. If the authors address these concerns, the manuscript will be significantly strengthened and provide a broader and more insightful understanding for readers.

Author Response

Dear Reviewer

We sincerely appreciate the time and effort you have dedicated to reviewing our manuscript. Your comments and suggestions have been invaluable in improving the quality and clarity of our work. Below, we provide a detailed, point-by-point response to address each of your comments and suggestions.

Thank you again for your thoughtful feedback.

Reviewer's Comment

The abstract mentions a unique application that cites empirical evidence, yet there is no in-depth evaluation of this app within the manuscript. A comparative analysis of its empirical results against the strengths and weaknesses of other applications would provide a more comprehensive perspective on successful mobile gaming applications.

Response: We appreciate your valuable suggestion regarding a comparative analysis. However, the primary goal of this scoping review was to systematically map the landscape of existing mobile gaming applications claiming cognitive benefits, assessing their scientific validation, accessibility, usability, and user engagement comprehensively. Given this aim, our intent was not to conduct an in-depth comparative analysis of individual app effectiveness or outcomes, but rather to broadly identify and evaluate whether apps meet established scientific and accessibility criteria. Conducting a detailed comparative analysis on a single application citing empirical evidence would diverge from our scoping review’s purpose, which is to provide an overarching evaluation framework rather than app-specific outcome comparisons. We have clearly noted this as a potential area for future detailed investigations involving comparative studies.

Table 1 reveals that none of the reviewed applications incorporate evidence-based cognitive training methodologies or assessment mechanisms to track cognitive progress. Given this, it would be beneficial to include a dedicated section—or at least a discussion paragraph—exploring cognitive enhancement methodologies from a healthcare perspective and evaluating the applications accordingly. All of the applications lack an explicit theoretical foundation. Offering recommendations for integrating established cognitive science principles into app development would significantly enhance the quality of the paper.

Response: Thank you for your valuable feedback. We have addressed this suggestion by including an additional paragraph in the Conclusion section that explicitly explores cognitive enhancement methodologies from a healthcare perspective.

The study primarily focuses on general usability features of the applications but does not sufficiently compare them against existing cognitive training interventions, whether digital or traditional. Strengthening the Related Work section by linking traditional cognitive training research to digital approaches would provide valuable context. While app store ratings and user reviews are included in the analysis, these metrics are susceptible to bias, such as promotional manipulation. The manuscript would benefit from incorporating usability studies to provide a more detailed assessment of user engagement.

Response: Thank you for your valuable suggestion regarding comparisons with traditional cognitive training interventions and potential biases inherent in app store ratings and user reviews. While an in-depth comparison between digital applications and traditional cognitive training falls outside the scope of our current review, the primary goal of this scoping review was to systematically map the landscape of existing mobile gaming applications claiming cognitive benefits by broadly assessing their scientific validation, accessibility, usability, and user engagement. Thus, rather than conducting a detailed comparative analysis, our intent was to evaluate whether existing apps meet established scientific and usability criteria at a high level. Nevertheless, we acknowledge these considerations as important limitations. Specifically, we recognize that promotional manipulation and biases in user ratings represent significant limitations, and we have explicitly noted these concerns. We suggest that future studies incorporate detailed usability analyses as important directions for future research.

Reviewer 3 Report

Comments and Suggestions for Authors

As the global population of older adults continues to grow, concerns regarding cognitive decline have increased. Cognitive training has emerged as a promising non-pharmacological intervention for maintaining cognitive health, and mobile gaming applications have been increasingly marketed as tools to enhance cognitive function in older adults. However, the scientific validity, accessibility, and effectiveness of these applications remain uncertain.

AUTHORS propose a scoping review aiming  to evaluate the current landscape of mobile gaming applications designed for cognitive enhancement in older adults.

A systematic search was conducted across the Apple App Store and Google Play Store using keywords related to cognitive training and aging. Eligible applications were those explicitly targeting older adults and designed to improve cognitive functioning. Applications were assessed based on their theoretical foundations, evidence-based claims, accessibility features, and targeted cognitive functions. A total of 14 cognitive training applications met the inclusion criteria. While all applications claimed to enhance cognitive function, only one cited empirical research to support its claims. Most applications lacked scientific validation and did not provide cognitive assessments or progress-tracking mechanisms. Accessibility features varied significantly, with vision-related accommodation being the most common, while auditory accessibility was largely absent. Additionally, the monetization models ranged from free access to subscription-based services, potentially limiting the affordability for some older users.

AUTHORS conclude that their review: (1) highlight critical gaps in the current cognitive gaming application market for older adults. (2) Despite their growing popularity, most applications lack empirical support, standardized cognitive training methodologies, and adequate accessibility features. (3) Developers should prioritize integrating evidence-based training principles, enhancing accessibility, and ensuring transparency in claims. (4) Future research should examine the real-world efficacy of these applications through longitudinal studies and user engagement analyses to determine their impact on cognitive health in aging populations.

This is a really interesting study.

The review touches on a hot and emerging topic.

Here are some comments for the authors:

1) Introduction. It does not follow the MDPI standard in citations, see “(Dinius et al., 2023; Yao et al., 2020)”. Furthermore, citations in groups should be avoided but it is necessary to go into detail on the single reference.

2) It could be useful to insert some narrow and essential key questions that the review intends to answer and then move on to the aims in the last part by connecting to what has already been written.

3) Impeccable methods. The section dedicated to the selections of Apps is very interesting and well developed

4) I believe that the results would benefit from a preliminary summary, a few lines to explain the section presentation strategy and the type of focus/analysis. Then you can start with the themes/sections you have identified.

5) Insert the conclusions that possibly must also contain suggestions for future developments.

Author Response

Dear Reviewer

We sincerely appreciate the time and effort you have dedicated to reviewing our manuscript. Your comments and suggestions have been invaluable in improving the quality and clarity of our work. Below, we provide a detailed, point-by-point response to address each of your comments and suggestions.

Thank you again for your thoughtful feedback.

1) Introduction. It does not follow the MDPI standard in citations, see “(Dinius et al., 2023; Yao et al., 2020)”. Furthermore, citations in groups should be avoided but it is necessary to go into detail on the single reference.

Response: Thank you for your suggestions. We have revised the reference formatting to align with the journal’s guidelines.

2) It could be useful to insert some narrow and essential key questions that the review intends to answer and then move on to the aims in the last part by connecting to what has already been written.

Response: Thank you for your valuable feedback. We have added some essential key questions at the end of the introduction.

3) Impeccable methods. The section dedicated to the selections of Apps is very interesting and well developed

Response: Thank you for your kind words.

4) I believe that the results would benefit from a preliminary summary, a few lines to explain the section presentation strategy and the type of focus/analysis. Then you can start with the themes/sections you have identified.

Response: Thank you for your suggestions. We have added a preliminary summary at the beginning of the Result section.

5) Insert the conclusions that possibly must also contain suggestions for future developments.

Response: We have added the required Conclusion section that contains suggestions for future development to our manuscript.

Round 2

Reviewer 1 Report

Comments and Suggestions for Authors

Many thanks for addressing all my comments.

Author Response

Dear Reviewer

We sincerely appreciate your dedicated time and effort in reviewing our manuscript. Thank you for your feedback.

Reviewer 2 Report

Comments and Suggestions for Authors

The recommendations for the authors have been thoroughly considered. Some limitations of the applications from a scientific perspective have been identified and added to the Discussion section. While the authors have clearly emphasized the context of the paper in the revision document, the details about the empirical evaluation have been indicated in Future Work.

Adding key questions at the end of the Introduction section is a valuable idea. However, the answers to these questions are currently dispersed across various sections of the manuscript. It would be more effective to consolidate these answers, for example, in the Discussion section, to coherently highlight the manuscript’s main contributions. Additionally, explicitly connecting the findings to the research questions would further enhance readability and clarity for the reader.

Author Response

Dear Reviewer

Thank you very much for reviewing our manuscript and for your valuable feedback. We have explicitly connected our findings to the research questions in the Discussion section to enhance readability and clarity for the reader.